# Discriminative Topic Modeling with Logistic LDA

**Iryna Korshunova**
Ghent University
`iryna.korshunova@ugent.be`

**Hanchen Xiong**
Twitter
`hxiong@twitter.com`

**Mateusz Fedoryszak**
Twitter
`mfedoryszak@twitter.com`

**Lucas Theis**
Twitter
`ltheis@twitter.com`

## Abstract

Despite many years of research into latent Dirichlet allocation (LDA), applying LDA to collections of non-categorical items is still challenging. Yet many problems with much richer data share a similar structure and could benefit from the vast literature on LDA. We propose *logistic LDA*, a novel discriminative variant of latent Dirichlet allocation which is easy to apply to arbitrary inputs. In particular, our model can easily be applied to groups of images, arbitrary text embeddings, and integrates well with deep neural networks. Although it is a discriminative model, we show that logistic LDA can learn from unlabeled data in an unsupervised manner by exploiting the group structure present in the data. In contrast to other recent topic models designed to handle arbitrary inputs, our model does not sacrifice the interpretability and principled motivation of LDA.

## 1 Introduction

Probabilistic topic models are powerful tools for discovering themes in large collections of items. Typically, these collections are assumed to be documents and the models assign topics to individual words. However, a growing number of real-world problems require assignment of topics to much richer sets of items. For example, we may want to assign topics to the tweets of an author on Twitter which contain multiple sentences as well as images, or to the images and websites stored in a board on Pinterest, or to the videos uploaded by a user on YouTube. These problems have in common that grouped items are likely to be thematically similar. We would like to exploit this dependency instead of categorizing items based on their content alone. Topic models provide a natural way to achieve this.

The most widely used topic model is latent Dirichlet allocation (LDA) [6]. With a few exceptions [7, 31], LDA and its variants, including supervised models [5, 19], are *generative*. They generally assume a multinomial distribution over words given topics, which limits their applicability to discrete tokens. While it is conceptually easy to extend LDA to continuous inputs [4], modeling the distribution of complex data such as images can be a difficult task on its own. Achieving low perplexity on images, for example, would require us to model many dependencies between pixels which are of little use for topic inference and would lead to inefficient models. On the other hand, a lot of progress has been made in accurately and efficiently assigning categories to images using *discriminative* models such as convolutional neural networks [18, 35].

In this work, our goal is to build a class of discriminative topic models capable of handling much richer items than words. At the same time, we would like to preserve LDA's extensibility and interpretability. In particular, group-level topic distributions and items should be independent given the item's topics, and topics and topic distributions should interact in an intuitive way. Our model

achieves these goals by discarding the generative part of LDA while maintaining the factorization of the conditional distribution over latent variables. By using neural networks to represent one of the factors, the model can deal with arbitrary input types. We call this model *logistic LDA* as its connection to LDA is analogous to the relationship between logistic regression and naive Bayes – textbook examples of discriminative and generative approaches [30].

A desirable property of generative models is that they can be trained in an unsupervised manner. In Section 6, we show that the grouping of items provides enough supervision to train logistic LDA in an otherwise unsupervised manner. We provide two approaches for training our model. In Section 5.1, we describe mean-field variational inference which can be used to train our model in an unsupervised, semi-supervised or supervised manner. In Section 5.2, we further describe an empirical risk minimization approach which can be used to optimize an arbitrary loss when labels are available.

When topic models are applied to documents, the topics associated with individual words are usually of little interest. In contrast, the topics of tweets on Twitter, pins on Pinterest, or videos on YouTube are of great interest. Therefore, we additionally introduce a new annotated dataset of tweets which allows us to explore model's ability to infer the topics of items. Our code and datasets are available at `github.com/lucastheis/logistic_lda`.

## 2 Related work

### 2.1 Latent Dirichlet allocation

LDA [6] is a latent variable model which relates observed words $x_{dn} \in \{1, \dots, V\}$ of document $d$ to latent topics $k_{dn} \in \{1, \dots, K\}$ and a distribution over topics $\pi_d$. It specifies the following generative process for a document:

1. Draw topic proportions $\pi_d \sim \text{Dir}(\alpha)$
2. For each word $x_{dn}$:
   (a) Draw a topic assignment $k_{dn} \sim \text{Cat}(\pi_d)$
   (b) Draw a word $x_{dn} \sim \text{Cat}(\beta^\top k_{dn})$

Here, we assume that topics and words are represented as vectors using a one-hot encoding, and $\beta$ is a $K \times V$ matrix where each row corresponds to a topic which parametrizes a categorical distribution over words. The matrix $\beta$ is either considered a parameter of the model or, more commonly, a latent variable with a Dirichlet prior over rows, i.e., $\beta_k \sim \text{Dir}(\eta)$ [6]. A graphical model corresponding to LDA is provided in Figure 1a.

Blei et al. [6] used mean-field variational inference to approximate the intractable posterior over latent variables $\pi_d$ and $k_{dn}$, resulting in closed-form coordinate ascent updates on a variational lower bound of the model's log-likelihood. Many other methods of inference have been explored, including Gibbs sampling [12], expectation propagation [28], and stochastic variants of variational inference [14].

It is worth noting that while LDA is most frequently used to model words, it can also be applied to collections of other items. For example, images can be viewed as collections of image patches, and by assigning each image patch to a discrete code word one could directly apply the model above [15, 39]. However, while for example clustering is a simple way to assign image patches to code words, it is unclear how to choose between different preprocessing approaches in a principled manner.

### 2.2 A zoo of topic models

Many topic models have built upon the ideas of LDA and extended it in various ways. One group of methods modifies LDA's assumptions regarding the forms of $p(\pi_d)$, $p(k_{dn}|\pi_d)$ and $p(x_{dn}|k_{dn})$ such that the model becomes more expressive. For example, Jo and Oh [16] modeled sentences instead of words. Blei and Jordan [4] applied LDA to images by extracting low-level features such as color and texture and using Gaussian distributions instead of categorical distributions. Other examples include correlated topic models [3], which replace the Dirichlet distribution over topics $p(\pi_d)$ with a logistic normal distribution, and hierarchical Dirichlet processes [37], which enable an unbounded number of topics. Srivastava and Sutton [36] pointed out that a major challenge with this approach is the need to rederive an inference algorithm with every change to the modeling assumptions. Thus, several papers

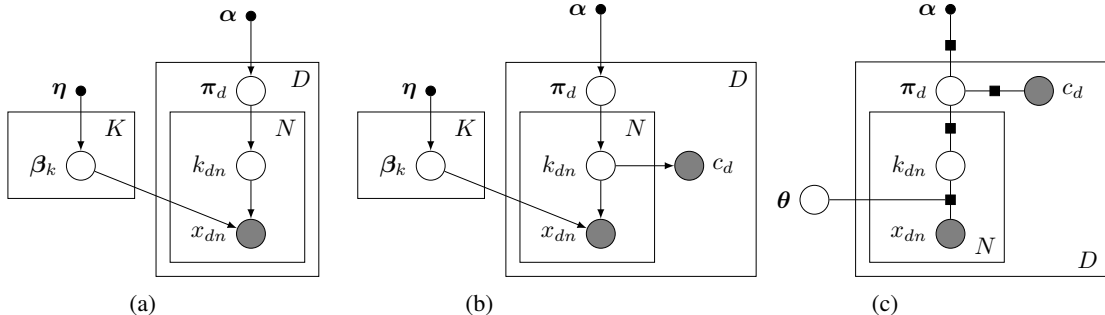

Figure 1: Graphical models for (a) LDA [6], (b) supervised LDA [5], and (c) logistic LDA. Gray circles indicate variables which are typically observed during training.

proposed to use neural variational inference but only tested their approaches on simple categorical items such as words [24, 25, 36].

Another direction of extending LDA is to incorporate extra information such as authorship [34], time [40], annotations [4], class labels or other features [27]. In this work, we are mainly interested in the inclusion of class labels as it covers a wide range of practical applications. In our model, in contrast to sLDA [5] and DiscLDA [19], labels interact with topic proportions instead of topics, and unlike in L-LDA [33], labels do not impose hard constraints on topic proportions.

A related area of research models documents without an explicit representation of topics, instead using more generic latent variables. These models are commonly implemented using neural networks and are sometimes referred to as *document models* to distinguish them from topic models which represent topics explicitly [24]. Examples of document models include Replicated Softmax [13], TopicRNN [10], NVDM [25], the Sigmoid Belief Document Model [29] and DocNADE [22].

Finally, non-probabilistic approaches to topic modeling employ heuristically designed loss functions. For example, Cao et al. [7] used a ranking loss to train an LDA inspired neural topic model.

## 3 An alternative view of LDA

In this section, we provide an alternative derivation of LDA as a special case of a broader class of models. Our goal is to derive a class of models which makes it easy to handle a variety of data modalities but which keeps the desirable inductive biases of LDA. In particular, topic distributions $\boldsymbol{\pi}_d$ and items $\boldsymbol{x}_{dn}$ should be independent given the items' topics $\boldsymbol{k}_{dn}$, and topics and topic distributions should interact in an intuitive way.

Instead of specifying a generative model as a directed network, we assume the factorization in Figure 1c and make the following three assumptions about the complete conditionals:

$$p(\boldsymbol{\pi}_d \mid \boldsymbol{k}_d) = \text{Dir}\left(\boldsymbol{\pi}_d; \boldsymbol{\alpha} + \sum_n \boldsymbol{k}_{dn}\right), \tag{1}$$

$$p(\boldsymbol{k}_{dn} \mid \boldsymbol{x}_{dn}, \boldsymbol{\pi}_d, \boldsymbol{\theta}) = \boldsymbol{k}_{dn}^\top \text{softmax}(g(\boldsymbol{x}_{dn}, \boldsymbol{\theta}) + \ln \boldsymbol{\pi}_d), \tag{2}$$

$$p(\boldsymbol{\theta} \mid \boldsymbol{x}, \boldsymbol{k}) \propto \exp\left(r(\boldsymbol{\theta}) + \sum_{dn} \boldsymbol{k}_{dn}^\top g(\boldsymbol{x}_{dn}, \boldsymbol{\theta})\right). \tag{3}$$

The first condition requires that the topic distribution is conditionally Dirichlet distributed, as in LDA. The second condition expresses how we would like to integrate information from $\boldsymbol{\pi}_d$ and $\boldsymbol{x}_{dn}$ to calculate beliefs over $\boldsymbol{k}_{dn}$. The function $g$ might be a neural network, in which case $\ln \boldsymbol{\pi}_d$ simply acts as an additional bias which is shared between grouped items. Finally, the third condition expresses what inference would look like if we knew the topics of all words. This inference step is akin to a classification problem with labels $\boldsymbol{k}_{dn}$, where $\exp r(\boldsymbol{\theta})$ acts as a prior and the remaining factors act as a likelihood.

In general, for an arbitrary set of conditional distributions, there is no guarantee that a corresponding joint distribution exists. The conditional distributions might be inconsistent, in which case no joint

distribution can satisfy all of them [1]. However, whenever a positive joint distribution exists we can use Brook's lemma [1] to find a form for the joint distribution, which in our case yields (Appendix B):

$$p(\boldsymbol{\pi}, \boldsymbol{k}, \boldsymbol{\theta} \mid \boldsymbol{x}) \propto \exp\left( (\boldsymbol{\alpha} - 1)^\top \sum_d \ln \boldsymbol{\pi}_d + \sum_{dn} \boldsymbol{k}_{dn}^\top \ln \boldsymbol{\pi}_d + \sum_{dn} \boldsymbol{k}_{dn}^\top g(\boldsymbol{x}_{dn}, \boldsymbol{\theta}) + r(\boldsymbol{\theta}) \right). \quad (4)$$

It is easy to verify that this distribution satisfies the constraints given by Equations 1 to 3. Furthermore, one can show that the posterior induced by LDA is a special case of Eq. 4, where

$$g(\boldsymbol{x}_{dn}, \boldsymbol{\beta}) = \ln \boldsymbol{\beta}\, \boldsymbol{x}_{dn}, \qquad\qquad r(\boldsymbol{\beta}) = (\boldsymbol{\eta} - 1)^\top \sum_k \ln \boldsymbol{\beta}_k, \qquad (5)$$

and $\boldsymbol{\beta}$ is constrained such that $\sum_j \boldsymbol{\beta}_{kj} = 1$ for all $k$ (Appendix A). However, Eq. 4 describes a larger class of models which share a very similar form of the posterior distribution.

The risk of relaxing assumptions is that it may prevent us from generalizing from data. An interesting question is therefore whether there are other choices of $g$ and $r$ which lead to useful models. In particular, does $g$ have to be a normalized log-likelihood as in LDA or can we lift this constraint? In the following, we answer this question positively.

### 3.1 Supervised extension

In many practical settings we have access to labels associated with the documents. It is not difficult to extend the model given by Equations 1 to 3 to the supervised case. However, there are multiple ways to do so. For instance, sLDA [5] assumes that a class variable $\boldsymbol{c}_d$ arises from the empirical frequencies of topic assignments $\boldsymbol{k}_{dn}$ within a document, as in Figure 1b. An alternative would be to have the class labels influence the topic proportions $\boldsymbol{\pi}_d$ instead. As a motivating example for the latter, consider the case where authors of documents belong to certain communities, each with a tendency to talk about different topics. Thus, even before observing any words of a new document, knowing the community provides us with information about the topic distribution $\boldsymbol{\pi}_d$. In sLDA, on the other hand, beliefs over topic distributions can only be influenced by labels once words $\boldsymbol{x}_{dn}$ have been observed.

Our proposed supervised extension therefore assumes Equations 2 and 3 together with the following conditionals:

$$p(\boldsymbol{c}_d \mid \boldsymbol{\pi}_d) = \mathrm{softmax}(\lambda \boldsymbol{c}_d^\top \ln \boldsymbol{\pi}_d), \qquad p(\boldsymbol{\pi}_d \mid \boldsymbol{k}_d, \boldsymbol{c}_d) = \mathrm{Dir}\left(\boldsymbol{\pi}_d; \boldsymbol{\alpha} + \textstyle\sum_n \boldsymbol{k}_{dn} + \lambda \boldsymbol{c}_d\right). \qquad (6)$$

Appendix B provides a derivation of the corresponding joint distribution, $p(\boldsymbol{\pi}, \boldsymbol{k}, \boldsymbol{c}, \boldsymbol{\theta} \mid \boldsymbol{x})$. Here, we assumed that the document label $\boldsymbol{c}_d$ is a $1 \times K$ one-hot vector and $\lambda$ is an extra scalar hyperparameter. Future work may want to explore the case where the number of classes is different from $K$ and $\lambda$ is replaced by a learnable matrix of weights.

## 4 Logistic LDA

Let us return to the question regarding the possible choices for $g$ and $r$ in Eq. 4. An interesting alternative to LDA is to require $\sum_k \boldsymbol{\beta}_{kj} = 1$. Instead of distributions over words, $\boldsymbol{\beta}$ in this case encodes a distribution over topics for each word and Eq. 3 turns into the posterior of a discriminative classifier rather than the posterior associated with a generative model over words. More generally, we can normalize $g$ such that it corresponds to a discriminative log-likelihood,

$$g(\boldsymbol{x}_{dn}, \boldsymbol{\theta}) = \ln \mathrm{softmax}\, f(\boldsymbol{x}_{dn}, \boldsymbol{\theta}), \qquad (7)$$

where $f$ outputs, for example, the $K$-dimensional logits of a neural network with parameters $\boldsymbol{\theta}$. Note that the conditional distribution over topics in Eq. 2 remains unchanged by this normalization.

Similar to how logistic regression and naive Bayes both implement linear classifiers but only naive Bayes makes assumptions about the distribution of inputs [30], our revised model shares the same conditional distribution over topics as LDA, but no longer make assumptions about the distribution of inputs $\boldsymbol{x}_{dn}$. We therefore refer to LDA-type models whose $g$ takes the form of Eq. 7 as *logistic LDA*.

Discriminative models typically require labels for training. But unlike other discriminative models, logistic LDA already receives a weak form of supervision through the partitioning of the dataset,

which encourages grouped items to be mapped to the same topics. Unfortunately, the assumptions of logistic LDA are still slightly too weak to produce useful beliefs. In particular, assigning all topics $\boldsymbol{k}_{dn}$ to the same value has high probability (Eq. 4). However, we found the following regularizer to be enough to encourage the use of all topics and to allow unsupervised training:

$$r(\boldsymbol{\theta}, \boldsymbol{x}) = \gamma \cdot \mathbf{1}^\top \ln \sum_{dn} \exp g(\boldsymbol{x}_{dn}, \boldsymbol{\theta}). \tag{8}$$

Here, we allow the regularizer to depend on the observed data, which otherwise does not affect the math in Section 3, and $\gamma$ controls the strength of the regularization. The regularizer effectively computes the average distribution of the item's topics as predicted by $g$ across the whole dataset and compares it to the uniform distribution. The proposed regularizer allows us to discover meaningful topics with logistic LDA in an unsupervised manner, although the particular form of the regularizer may not be crucial.

To make the regularizer more amenable to stochastic approximation, we lower-bound it as follows:

$$r(\boldsymbol{\theta}, \boldsymbol{x}) \geq \gamma \sum_k \sum_{dn} r_{dnk} \ln \frac{\exp g_k(\boldsymbol{x}_{dn}, \boldsymbol{\theta})}{r_{dnk}} \qquad r_{dnk} = \frac{\exp g_k(\boldsymbol{x}_{dn}, \boldsymbol{\theta})}{\sum_{dn} \exp g_k(\boldsymbol{x}_{dn}, \boldsymbol{\theta})} \tag{9}$$

For fixed $r_{dnk}$ evaluated at $\boldsymbol{\theta}$, the lower bound has the same gradient as $r(\boldsymbol{\theta}, \boldsymbol{x})$. In practice, we are further approximating the denominator of $r_{dnk}$ with a running average, yielding an estimate $\hat{r}_{dnk}$ (see Appendix D for details).

## 5  Training and inference

### 5.1  Mean-field variational inference

We approximate the intractable posterior (Eq. 4) with a factorial distribution via mean-field variational inference, that is, by minimizing the Kullback-Leibler (KL) divergence

$$D_{\mathrm{KL}} \left[ q(\boldsymbol{\theta}) \left( \prod_d q(\boldsymbol{c}_d) \right) \left( \prod_d q(\boldsymbol{\pi}_d) \right) \left( \prod_{dn} q(\boldsymbol{k}_{dn}) \right) \, || \, p(\boldsymbol{\pi}, \boldsymbol{k}, \boldsymbol{c}, \boldsymbol{\theta} \mid \boldsymbol{x}) \right] \tag{10}$$

with respect to the distributions $q$. Assuming for now that the distribution over $\boldsymbol{\theta}$ is concentrated on a point estimate, i.e., $q(\boldsymbol{\theta}; \hat{\boldsymbol{\theta}}) = \delta(\boldsymbol{\theta} - \hat{\boldsymbol{\theta}})$, we can derive the following coordinate descent updates for the variational parameters (see Appendix C for more details):

$$q(\boldsymbol{c}_d) = \boldsymbol{c}_d^\top \hat{\boldsymbol{p}}_d \qquad\qquad \hat{\boldsymbol{p}}_d = \mathrm{softmax}\left(\lambda \psi(\hat{\boldsymbol{\alpha}}_d)\right) \tag{11}$$

$$q(\boldsymbol{\pi}_d) = \mathrm{Dir}(\boldsymbol{\pi}_d; \hat{\boldsymbol{\alpha}}_d) \qquad\qquad \hat{\boldsymbol{\alpha}}_d = \boldsymbol{\alpha} + \sum_n \hat{\boldsymbol{p}}_{dn} + \lambda \hat{\boldsymbol{p}}_d \tag{12}$$

$$q(\boldsymbol{k}_{dn}) = \boldsymbol{k}_{dn}^\top \hat{\boldsymbol{p}}_{dn} \qquad\qquad \hat{\boldsymbol{p}}_{dn} = \mathrm{softmax}\left(f(\boldsymbol{x}_{dn}, \hat{\boldsymbol{\theta}}) + \psi(\hat{\boldsymbol{\alpha}}_d)\right) \tag{13}$$

Here, $\psi$ is the digamma function and $f$ are the logits of a neural network with parameters $\hat{\boldsymbol{\theta}}$. From Eq. 13, we see that topic predictions for a word $\boldsymbol{x}_{dn}$ are computed based on biased logits. The bias $\psi(\hat{\boldsymbol{\alpha}}_d)$ aggregates information across all items of a group (e.g., words of a document), thus providing context for individual predictions.

Iterating Equations 11 to 13 in arbitrary order implements a valid inference algorithm for any fixed $\hat{\boldsymbol{\theta}}$. Note that inference does not depend on the regularizer. To optimize the neural network's weights $\hat{\boldsymbol{\theta}}$, we fix the values of the variational parameters $\hat{\boldsymbol{p}}_{dn}$ and regularization terms $\hat{\boldsymbol{r}}_{dn}$. We then optimize the KL divergence in Eq. 10 with respect to $\hat{\boldsymbol{\theta}}$, which amounts to minimizing the following cross-entropy loss:

$$\ell(\hat{\boldsymbol{\theta}}) \approx - \sum_{dn} (\hat{\boldsymbol{p}}_{dn} + \gamma \cdot \hat{\boldsymbol{r}}_{dn})^\top g(\boldsymbol{x}_{dn}, \hat{\boldsymbol{\theta}}). \tag{14}$$

This corresponds to a classification problem with soft labels $\hat{\boldsymbol{p}}_{dn} + \gamma \cdot \hat{\boldsymbol{r}}_{dn}$. Intuitively, $\hat{\boldsymbol{p}}_{dn}$ tries to align predictions for grouped items, while $\hat{\boldsymbol{r}}_{dn}$ tries to ensure that each topic is predicted at least some of the time.

Thus far, we presented a general way of training and inference in logistic LDA, where we assumed $c_d$ to be a latent variable. If class labels are observed for some or all of the documents, we can replace $\hat{p}_d$ with $c_d$ during training. This makes the method suitable for unsupervised, semi-supervised and supervised learning. For supervised training with labels, we further developed the discriminative training procedure below.

## 5.2 Discriminative training

Decision tasks are associated with loss functions, e.g., in classification we often care about accuracy. Variational inference, however, only approximates general properties of a posterior while ignoring the task in which the approximation is going to be used, leading to suboptimal results [20]. When enough labels are available and classification is the goal, we therefore propose to directly optimize parameters $\hat{\theta}$ with respect to an empirical loss instead of the KL divergence above, e.g., a cross-entropy loss:

$$\ell(\hat{\boldsymbol{\theta}}) = -\sum_d \boldsymbol{c}_d^\top \ln \hat{\boldsymbol{p}}_d \tag{15}$$

To see why this is possible, note that each update in Equations 11 to 13 leading to $\hat{p}_d$ is a differentiable operation. In effect, we are unrolling the mean-field updates and treat them like layers of a sophisticated neural network. This strategy has been succesfully used before, for example, to improve performance of CRFs in semantic segmentation [41]. Unrolling mean-field updates leads to the training procedure given in Algorithm 1. The algorithm reveals that training and inference can be implemented easily even when the derivations needed to arrive at this algorithm may have seemed complex.

Algorithm 1 requires processing of all words of a document in each iteration. In Appendix D, we discuss highly scalable implementations of variational training and inference which only require looking at a single item at a time. This is useful in settings with many items or where items are more complex than words.

---

**Algorithm 1** Single step of discriminative training for a collection $\{\boldsymbol{x}_{dn}\}_{n=1}^{N_d}$ with class label $\boldsymbol{c}_d$.

---

**Require:** $\{\boldsymbol{x}_{dn}\}_{n=1}^{N_d}, \boldsymbol{c}_d$

$\quad \hat{\boldsymbol{\alpha}}_d \leftarrow \boldsymbol{\alpha}$
$\quad \hat{\boldsymbol{p}}_d \leftarrow \mathbf{1}/K$ % uniform initial beliefs over $K$ classes

$\quad$ **for** $i \leftarrow 1$ to $N_{\text{iter}}$ **do**
$\quad\quad$ **for** $n \leftarrow 1$ to $N_d$ **do**
$\quad\quad\quad \hat{\boldsymbol{p}}_{dn} \leftarrow \text{softmax}\left(f(\boldsymbol{x}_{dn}, \hat{\boldsymbol{\theta}}) + \psi(\hat{\boldsymbol{\alpha}}_d)\right)$ % Eq. 13; $f$ outputs $K$ logits of a neural net
$\quad\quad$ **end for**
$\quad\quad \hat{\boldsymbol{\alpha}}_d \leftarrow \boldsymbol{\alpha} + \sum_n \hat{\boldsymbol{p}}_{dn} + \lambda \hat{\boldsymbol{p}}_d$ % Eq. 12
$\quad\quad \hat{\boldsymbol{p}}_d \leftarrow \text{softmax}\left(\lambda \psi(\hat{\boldsymbol{\alpha}}_d)\right)$ % Eq. 11
$\quad$ **end for**

$\quad \hat{\boldsymbol{\theta}} \leftarrow \hat{\boldsymbol{\theta}} - \varepsilon \nabla_\theta \text{cross\_entropy}(\boldsymbol{c}_d, \hat{\boldsymbol{p}}_d)$

---

## 6 Experiments

While a lot of research has been done on models related to LDA, benchmarks have almost exclusively focused on either document classification or on a generative model's perplexity. However, here we are not only interested in logistic LDA's ability to discover the topics of documents but also those of individual items, as well as its ability to handle arbitrary types of inputs. We therefore explore two new benchmarks. First, we are going to look at a model's ability to discover the topics of tweets. Second, we are going to evaluate a model's ability to predict the categories of boards on Pinterest based on images. To connect with the literature on topic models and document classifiers, we are going to show that logistic LDA can also work well when applied to the task of document classification. Finally, we demonstrate that logistic LDA can recover meaningful topics from Pinterest and 20-Newsgroups in an unsupervised manner.

## 6.1 Topic classification on Twitter

We collected two sets of tweets. The first set contains 1.45 million tweets of 66,455 authors. Authors were clustered based on their follower graph, assigning each author to one or more communities. The clusters were subsequently manually annotated based on the content of typical tweets in the community. The community label thus provides us with an idea of the content an author is likely to produce. The second dataset contains 3.14 million tweets of 18,765 authors but no author labels. Instead, 18,864 tweets were manually annotated with one out of 300 topics from the same taxonomy used to annotate communities. We split the first dataset into training (70%), validation (10%), and test (20%) sets such that tweets of each author were only contained in one of the sets. The second dataset was only used for evaluation. Due to the smaller size of the second dataset, we here used 10-fold cross-validation to estimate the performance of all models.

During training, we used community labels to influence the distribution over topic weights via $c_d$. Where authors belonged to multiple communities, a label was chosen at random (i.e., the label is noisy). Labels were not used during inference but only during training. Tweets were embedded by averaging 300-dimensional skip-gram embeddings of words [26]. Logistic LDA applied a shallow MLP on top of these embeddings and was trained using a stochastic approximation to mean-field variational inference (Section 5.1). As baselines, we tried LDA as well as training an MLP to predict the community labels directly. To predict the community of authors with an MLP, we used majority voting across the predictions for their tweets. The main difference between majority voting and logistic LDA is that the latter is able to reevaluate predictions for tweets based on other tweets of an author. For LDA, we extended the open source implementation of Theis and Hoffman [38] to depend on the label in the same manner as logistic LDA. That is, the label biases topic proportions as in Figure 1c. The words in all tweets of an author were combined to form a document, and the 100,000 most frequent words of the corpus formed the vocabulary of LDA. To predict the topics of tweets, we averaged LDA's beliefs over the topics of words contained in the tweet, $\sum_m q(\boldsymbol{k}_{dm})/M$.

Table 1 shows that logistic LDA is able to improve the predictions of a purely discriminatively trained neural network for both author- and tweet-level categories. More principled inference allows it to improve the accuracy of predictions of the communities of authors, while integrating information from other tweets allows it to improve the prediction of a tweet's topic. LDA performed worse on both tasks. We note that labels of the dataset are noisy and difficult to predict even for humans, hence the relatively low accuracy numbers.

Table 1: Accuracy of prediction of annotations at the author and tweet level. Authors were annotated with communities, tweets with topics. LDA here refers to a supervised generative model.

| Model | Author | Tweet |
|---|---|---|
| MLP (individual) | 26.6% | 32.4% |
| MLP (majority) | 35.0% | n/a |
| LDA | 33.1% | 25.4% |
| Logistic LDA | **38.7%** | **35.6%** |

Table 2: Accuracy on 20-Newsgroups.

| Model | Test accuracy |
|---|---|
| SVM [8] | 82.9% |
| LSTM [9] | 82.0% |
| SA-LSTM [9] | 84.4% |
| oh-2LSTMp [17] | **86.5%** |
| Logistic LDA | 84.4% |

## 6.2 Image categorization on Pinterest

To illustrate how logistic LDA can be used with images, we apply it to Pinterest data of boards and pins. In LDA's terms, every board would correspond to a document and every pin – an image pinned to a board – to a word or an item. For our purpose, we used a subset of the Pinterest dataset of Geng et al. [11], which we describe in Appendix F. It should be noted, however, that the dataset contains only board labels. Thus, without pin labels, we are not able to perform the same in-depth analysis as in the previous section.

As in case of Twitter data, we trained logistic LDA with our stochastic variational inference procedure. For comparison, we trained an MLP to predict the labels of individual pins, where each pin was labeled with the category of its board. For both models, we used image embeddings from a MobileNetV2 [35] as inputs, and tuned the hyperparameters on a validation set.

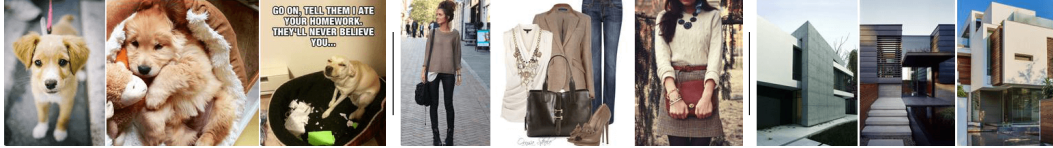

Figure 2: Top-3 images assigned to three different topics discovered by logistic LDA in an unsupervised manner (dogs, fashion, architecture).

Test accuracy when predicting board labels for logistic LDA and MLP was 82.5% and 81.3%, respectively. For MLP, this score was obtained using majority voting across pins to compute board predictions. We further trained logistic LDA in an unsupervised manner. Image embeddings are subsequently mapped to topics using the trained neural network. We find that logistic LDA is able to learn coherent topics in an unsupervised manner. Examples topics are visualized in Figure 2. Further details and more results are provided in Appendix G and H.

## 6.3 Document classification

We apply logistic LDA with discrimintive training (Section 5.2) to the standard benchmark problem of document classification on the 20-Newsgroups dataset [21]. 20-Newsgroups comprises of around 18,000 posts partitioned almost evenly among 20 topics. While various versions of this dataset exist, we used the preprocessed version of Cardoso-Cachopo [8] so our results can be compared to the ones from LSTM-based classifiers [9, 17]. More details on this dataset are given in Appendix F.

We trained logistic LDA with words represented as 300-dimensional GloVe embeddings [32]. The hyperparameters were selected based on a 15% split from the training data and are listed in Appendix E. Results of these experiments are given in Table 2. As a baseline, we include an SVM model trained on tf-idf document vectors [8]. We also compare logistic LDA to an LSTM model for document classification [9], which owes its poorer performance to instable training and difficulties when dealing with long texts. These issues can be overcome by starting from a pretrained model or by using more intricate architectures. SA-LSTM [9] adopts the former approach, while oh-2LSTMp [17] implements the latter. To our knowledge, oh-2LSTMp holds the state-of-the-art results on 20-Newsgroups. While logistic LDA does not surpass oh-2LSTMp on this task, its performance compares favourably to other more complex models. Remarkably, it achieves the same results as SA-LSTM – an LSTM classifier initialized with a pretrained sequence autoencoder [9]. It is worth noting that logistic LDA uses generic word embeddings, is a lightweight model which requires hours instead of days to train, and provides explicit representations of topics.

In this accuracy-driven benchmark, it is interesting to look at the performance of a supervised logistic LDA trained with the loss-insensitive objective for $\hat{\theta}$ as described in Section 5.1. Our best accuracy with this method was 82.2% – a significantly worse result compared to 84.4% achieved with logistic LDA that used a cross-entropy loss in Eq. 15 when optimizing for $\hat{\theta}$. This confirms the usefulness of optimizing inference for the task at hand [20].

The benefit of mean-field variational inference (Section 5.1) is that it allows to train logistic LDA in an unsupervised manner. In this case, we find that the model is able to discover topics such as the ones given in Table 3. Qualitative comparisons with Srivastava and Sutton [36] together with NPMI topic coherence scores [23] of multiple models can be found in Appendix I.

Table 3: Examples of topics discovered by unsupervised logistic LDA represented by top-10 words

| | |
|---|---|
| **1** | bmw, motor, car, honda, motorcycle, auto, mg, engine, ford, bike |
| **2** | christianity, prophet, atheist, religion, holy, scripture, biblical, catholic, homosexual, religious, atheist |
| **3** | spacecraft, orbit, probe, ship, satellite, rocket, surface, shipping, moon, launch |
| **4** | user, computer, microsoft, monitor,programmer, electronic, processing, data, app, systems |
| **5** | congress, administration, economic, accord, trade, criminal, seriously, fight, responsible, future |

# 7 Discussion and conclusion

We presented logistic LDA, a neural topic model that preserves most of LDA's inductive biases while giving up its generative component in favour of a discriminative approach, making it easier to apply to a wide range of data modalities.

In this paper we only scratched the surface of what may be possible with discriminative variants of LDA. Many inference techniques have been developed for LDA and could be applied to logistic LDA. For example, mean-field variational inference is known to be prone to local optima but trust-region methods are able to get around them [38]. We only trained fairly simple neural networks on precomputed embeddings. An interesting question will be whether much deeper neural networks can be trained using only weak supervision in the form of grouped items.

Interestingly, logistic LDA would not be considered a discriminative model if we follow the definition of Bishop and Lasserre [2]. According to this definition, a discriminative model's joint distribution over inputs $x$, labels $c$, and model parameters $\theta$ factorizes as $p(c \mid x, \theta)p(\theta)p(x)$. Logistic LDA, on the other hand, only admits the factorization $p(c, \theta \mid x)p(x)$. Both have in common that the choice of marginal $p(x)$ has no influence on inference, unlike in a generative model.

### Acknowledgments

This work was done while Iryna Korshunova was an intern at Twitter, London. We thank Ferenc Huszár, Jonas Degrave, Guy Hugot-Derville, Francisco Ruiz, and Dawen Liang for helpful discussions and feedback on the manuscript.

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
