[Supplementary Material]

## Appendix A

In the following, we will provide LDA's complete conditional distibutions for $\boldsymbol{\pi}_d$, $\boldsymbol{k}_{dn}$ and $\boldsymbol{\beta}$, but firstly, let us remind ourselves of LDA's generative process:

1. Draw topics proportions $\boldsymbol{\pi}_d \sim \text{Dir}(\boldsymbol{\alpha})$
2. For each word $\boldsymbol{x}_{dn}$:
    (a) Draw a topic assignment $\boldsymbol{k}_{dn} \sim \text{Cat}(\boldsymbol{\pi}_d)$
    (b) Draw a word $\boldsymbol{x}_{dn} \sim \text{Cat}(\boldsymbol{\beta}^\top \boldsymbol{k}_{dn})$

Here, $\boldsymbol{\alpha}$ is a $K$-dimensional vector of Dirichlet concentration parameters, where $K$ is the number of topics, $\boldsymbol{\pi}_d$ is $K \times 1$ vector of topic proportions, and $\boldsymbol{\beta}$ is a $K \times V$ matrix where $V$ is the number of words in a vocabulary. Every $\boldsymbol{\beta}_{ij}$ specifies the conditional probability of a word $j$ given a topic $i$, s.t. $\sum_j \boldsymbol{\beta}_{ij} = 1$. We will use one-hot encoding for words and topics such that $\boldsymbol{x}_{dn}$ is a $V \times 1$ vector, and $\boldsymbol{k}_{dn}$ is a $K \times 1$ vector. We will drop the indices of documents and words, $d$ and $n$, to denote the collection of variables. For example, $\boldsymbol{x}$ is a collection of words across the dataset.

**1.** The conditional distribution $p(\boldsymbol{\pi}_d \mid \boldsymbol{k}_d)$ is the posterior of a Dirichlet-categorical model, and it can be shown to take the form:

$$p(\boldsymbol{\pi}_d \mid \boldsymbol{k}_d) = \text{Dir}\left(\boldsymbol{\pi}_d; \boldsymbol{\alpha} + \sum_n \boldsymbol{k}_{dn}\right)$$

$$\propto \exp\left(\left(\boldsymbol{\alpha} + \sum_n \boldsymbol{k}_{dn} - 1\right)^\top \ln \boldsymbol{\pi}_d\right). \tag{1}$$

**2.** The conditional distribution over topic assignments is given by:

$$
\begin{aligned}
p(\boldsymbol{k}_{dn} \mid \boldsymbol{x}_{dn}, \boldsymbol{\pi}_d, \boldsymbol{\beta}) &\propto p(\boldsymbol{x}_{dn} \mid \boldsymbol{\pi}_d, \boldsymbol{k}_{dn}, \boldsymbol{\beta}) p(\boldsymbol{k}_d \mid \boldsymbol{\pi}_d, \boldsymbol{\beta}) \\
&= p(\boldsymbol{x}_{dn} \mid \boldsymbol{k}_{dn}, \boldsymbol{\beta}) p(\boldsymbol{k}_{dn} \mid \boldsymbol{\pi}_d) \\
&= \left(\boldsymbol{k}_{dn}^\top \boldsymbol{\beta} \boldsymbol{x}_{dn}\right)\left(\boldsymbol{k}_{dn}^\top \boldsymbol{\pi}_d\right) \\
&= \exp\left(\boldsymbol{k}_{dn}^\top \ln \boldsymbol{\beta} \boldsymbol{x}_{dn} + \boldsymbol{k}_{dn}^\top \ln \boldsymbol{\pi}_d\right).
\end{aligned} \tag{2}
$$

**3.** Finally, we derive the posterior distribution of $\boldsymbol{\beta}$ assuming a Dirichlet prior over the rows of $\boldsymbol{\beta}$, i.e., $p(\boldsymbol{\beta}_k) = \text{Dir}(\boldsymbol{\eta})$ for every $k = 1, \ldots, K$:

$$
\begin{aligned}
p(\boldsymbol{\beta} \mid \boldsymbol{x}, \boldsymbol{k}) &\propto p(\boldsymbol{x} \mid \boldsymbol{\beta}, \boldsymbol{k}) p(\boldsymbol{\beta} \mid \boldsymbol{k}) \\
&= \left(\prod_{dn} \boldsymbol{k}_{dn}^\top \boldsymbol{\beta} \boldsymbol{x}_{dn}\right)\left(\prod_k p(\boldsymbol{\beta}_k)\right) \\
&= \left(\prod_{dn} \boldsymbol{k}_{dn}^\top \boldsymbol{\beta} \boldsymbol{x}_{dn}\right)\left(\prod_k \frac{1}{\mathcal{B}(\boldsymbol{\eta})} \prod_i \boldsymbol{\beta}_{ki}^{\eta_i - 1}\right) \\
&= \exp\left(\sum_{dn} \boldsymbol{k}_{dn}^\top \ln \boldsymbol{\beta} \boldsymbol{x}_{dn} + (\boldsymbol{\eta} - 1)^\top \sum_k \ln \boldsymbol{\beta}_k - K \ln \mathcal{B}(\boldsymbol{\eta})\right).
\end{aligned} \tag{3}
$$

Here, $\mathcal{B}$ is a multivariate beta function.

## Appendix B

Given a set of assumptions about the conditional distributions of logistic LDA, we would like to derive the joint conditional distribution over $\boldsymbol{\pi}$, $\boldsymbol{k}$ and $\boldsymbol{\theta}$. For that, we can use Brook's lemma [1]:

**Brook's Lemma.** *Let $p(\boldsymbol{y}) > 0$ for all $\boldsymbol{y}$. Then for any $\boldsymbol{y}$ and $\boldsymbol{y}'$ we have*

$$\frac{p(\boldsymbol{y})}{p(\boldsymbol{y}')} = \prod_i \frac{p(\boldsymbol{y}_i \mid \boldsymbol{y}_1, \ldots, \boldsymbol{y}_{i-1}, \boldsymbol{y}'_{i+1}, \ldots, \boldsymbol{y}'_M)}{p(\boldsymbol{y}'_i \mid \boldsymbol{y}_1, \ldots, \boldsymbol{y}_{i-1}, \boldsymbol{y}'_{i+1}, \ldots, \boldsymbol{y}'_M)}.$$

Logistic LDA specifies the following conditionals:

$$p(\boldsymbol{\pi}_d \mid \boldsymbol{k}_d) \propto \exp\left(\left(\boldsymbol{\alpha} + \sum_n \boldsymbol{k}_{dn} - 1\right)^\top \ln \boldsymbol{\pi}_d\right), \tag{4}$$

$$p(\boldsymbol{k}_{dn} \mid \boldsymbol{x}_{dn}, \boldsymbol{\pi}_d, \boldsymbol{\theta}) \propto \exp\left(\boldsymbol{k}_{dn}^\top g(\boldsymbol{x}_{dn}, \boldsymbol{\theta}) + \boldsymbol{k}_{dn}^\top \ln \boldsymbol{\pi}_d\right), \tag{5}$$

$$p(\boldsymbol{\theta} \mid \boldsymbol{x}, \boldsymbol{k}) \propto \exp\left(r(\boldsymbol{\theta}) + \sum_{dn} \boldsymbol{k}_{dn}^\top g(\boldsymbol{x}_{dn}, \boldsymbol{\theta})\right). \tag{6}$$

Using Brook's lemma:

$$p(\boldsymbol{\pi}, \boldsymbol{k}, \boldsymbol{\theta} \mid \boldsymbol{x}) \propto \frac{p(\boldsymbol{\pi} \mid \boldsymbol{k}')p(\boldsymbol{k} \mid \boldsymbol{\pi}, \boldsymbol{\theta}', \boldsymbol{x})p(\boldsymbol{\theta} \mid \boldsymbol{k}, \boldsymbol{x})}{p(\boldsymbol{\pi}' \mid \boldsymbol{k}')p(\boldsymbol{k}' \mid \boldsymbol{\pi}, \boldsymbol{\theta}', \boldsymbol{x})p(\boldsymbol{\theta}' \mid \boldsymbol{k}, \boldsymbol{x})}$$

$$\propto \exp\left((\boldsymbol{\alpha} - 1)^\top \sum_d \ln \boldsymbol{\pi}_d + \sum_{dn} \boldsymbol{k}_{dn}^\top \ln \boldsymbol{\pi}_d + \sum_{dn} \boldsymbol{k}_{dn}^\top g(\boldsymbol{x}_{dn}, \boldsymbol{\theta}) + r(\boldsymbol{\theta})\right) \tag{7}$$

This is easiest to see if we chose $\boldsymbol{\theta}'$ such that $g(\boldsymbol{x}_{dn}, \boldsymbol{\theta}') = 0$, which Brook's lemma allows us to do. In case of logistic LDA with labels, we have a slightly different set of conditionals:

$$p(\boldsymbol{c}_d \mid \boldsymbol{\pi}_d) \propto \exp\left(\lambda \boldsymbol{c}_d^\top \ln \boldsymbol{\pi}_d\right), \tag{8}$$

$$p(\boldsymbol{\pi}_d \mid \boldsymbol{k}_d, \boldsymbol{c}_d) \propto \exp\left(\left(\boldsymbol{\alpha} + \sum_n \boldsymbol{k}_{dn} - 1 + \lambda \boldsymbol{c}_d\right)^\top \ln \boldsymbol{\pi}_d\right), \tag{9}$$

$$p(\boldsymbol{k}_{dn} \mid \boldsymbol{x}_{dn}, \boldsymbol{\pi}_d, \boldsymbol{\theta}) \propto \exp\left(\boldsymbol{k}_{dn}^\top g(\boldsymbol{x}_{dn}, \boldsymbol{\theta}) + \boldsymbol{k}_{dn}^\top \ln \boldsymbol{\pi}_d\right), \tag{10}$$

$$p(\boldsymbol{\theta} \mid \boldsymbol{x}, \boldsymbol{k}) \propto \exp\left(r(\boldsymbol{\theta}) + \sum_{dn} \boldsymbol{k}_{dn}^\top g(\boldsymbol{x}_{dn}, \boldsymbol{\theta})\right). \tag{11}$$

Similarly to Eq. 7, we can derive a joint distribution as:

$$p(\boldsymbol{\pi}, \boldsymbol{k}, \boldsymbol{c}, \boldsymbol{\theta} \mid \boldsymbol{x}) \propto \frac{p(\boldsymbol{\pi} \mid \boldsymbol{k}')p(\boldsymbol{k} \mid \boldsymbol{\pi}, \boldsymbol{\theta}', \boldsymbol{x})p(\boldsymbol{c} \mid \boldsymbol{\pi})p(\boldsymbol{\theta} \mid \boldsymbol{k}, \boldsymbol{x})}{p(\boldsymbol{\pi}' \mid \boldsymbol{k}')p(\boldsymbol{k}' \mid \boldsymbol{\pi}, \boldsymbol{\theta}', \boldsymbol{x})p(\boldsymbol{c}' \mid \boldsymbol{\pi})p(\boldsymbol{\theta}' \mid \boldsymbol{k}, \boldsymbol{x})}$$

$$\propto \exp\left((\boldsymbol{\alpha} - 1)^\top \sum_d \ln \boldsymbol{\pi}_d + \sum_{dn} \boldsymbol{k}_{dn}^\top \ln \boldsymbol{\pi}_d + \lambda \sum_d \boldsymbol{c}_d^\top \ln \boldsymbol{\pi}_d + \sum_{dn} \boldsymbol{k}_{dn}^\top g(\boldsymbol{x}_{dn}, \boldsymbol{\theta}) + r(\boldsymbol{\theta})\right)$$

## Appendix C

We assume the following parameterizations of the variational distributions:

$$\begin{aligned} q(\boldsymbol{c}_d) &= \boldsymbol{c}_d^\top \hat{\boldsymbol{p}}_d \\ q(\boldsymbol{\pi}_d) &= \text{Dir}(\boldsymbol{\pi}_d; \hat{\boldsymbol{\alpha}}_d) \\ q(\boldsymbol{k}_{dn}) &= \boldsymbol{k}_{dn}^\top \hat{\boldsymbol{p}}_{dn} \end{aligned} \tag{12}$$

For now, we will fix $\boldsymbol{\theta}$ to its point estimate $\hat{\boldsymbol{\theta}}$ and derive the optimal variational densities for other latent variables. Our derivation is based on the coordinate ascent variational inference algorithm (CAVI) [2].

**1.** Optimal variational density for $\boldsymbol{\pi}_d$:

$$q^*(\boldsymbol{\pi}_d) \propto \exp\left(\mathbb{E}_{q(\boldsymbol{k}_{dn}), q(\boldsymbol{c}_d)}\left[\sum_n \boldsymbol{k}_{dn}^\top \ln \boldsymbol{\pi}_d + (\boldsymbol{\alpha} - 1)^\top \ln \boldsymbol{\pi}_d + \lambda \boldsymbol{c}_d^\top \ln \boldsymbol{\pi}_d\right]\right)$$

$$= \exp\left(\left(\boldsymbol{\alpha} + \sum_n \mathbb{E}_{q(\boldsymbol{k}_{dn})}\boldsymbol{k}_{dn} + \lambda \mathbb{E}_{q(\boldsymbol{c}_d)}\boldsymbol{c}_d - 1\right)^\top \ln \boldsymbol{\pi}_d\right) \tag{13}$$

$$= \exp\left(\left(\boldsymbol{\alpha} + \sum_n \hat{\boldsymbol{p}}_{dn} + \lambda \hat{\boldsymbol{p}}_d - 1\right)^\top \ln \boldsymbol{\pi}_d\right)$$

Thus, the CAVI update for the variational parameter $\hat{\boldsymbol{\alpha}}$ is:

$$\hat{\boldsymbol{\alpha}} = \boldsymbol{\alpha} + \sum_n \hat{\boldsymbol{p}}_{dn} + \lambda \hat{\boldsymbol{p}}_d. \tag{14}$$

**2.** Optimal variational density for $\boldsymbol{k}_{dn}$:

$$
\begin{aligned}
q(\boldsymbol{k}_{dn}) &\propto \exp\left(\mathbb{E}_{\boldsymbol{\theta}\sim q(\boldsymbol{\theta}),\boldsymbol{\pi}_d\sim q(\boldsymbol{\pi}_d)}\left[\boldsymbol{k}_{dn}^\top \ln \boldsymbol{\pi}_d + \boldsymbol{k}_{dn}^\top g(\boldsymbol{x}_{dn},\boldsymbol{\theta})\right]\right) \\
&= \exp\left(\boldsymbol{k}_{dn}^\top \mathbb{E}_{q(\boldsymbol{\pi}_d)}\left[\ln \boldsymbol{\pi}_d\right] + \boldsymbol{k}_{dn}^\top g(\boldsymbol{x}_{dn},\hat{\boldsymbol{\theta}})\right) \\
&= \exp\left(\boldsymbol{k}_{dn}^\top(\psi(\hat{\boldsymbol{\alpha}}_d) - \psi\left(\sum_k \hat{\alpha}_{dk}\right) + g(\boldsymbol{x}_{dn},\hat{\boldsymbol{\theta}}))\right)
\end{aligned}
\tag{15}
$$

We now see that $\ln \hat{\boldsymbol{p}}_{dn} \propto \exp(\psi(\hat{\boldsymbol{\alpha}}_d) - \psi(\sum_k \hat{\alpha}_{dk}) + g(\boldsymbol{x}_{dn},\hat{\boldsymbol{\theta}}))$. After renormalizing:

$$\hat{\boldsymbol{p}}_{dn} = \frac{\exp(\psi(\hat{\boldsymbol{\alpha}}_d) + g(\boldsymbol{x}_{dn},\hat{\boldsymbol{\theta}}))}{\sum_k \exp(\psi(\hat{\alpha}_{dk}) + g_k(\boldsymbol{x}_{dn},\hat{\boldsymbol{\theta}}))}. \tag{16}$$

**3.** Optimal variational density for $\boldsymbol{c}_d$:

$$q(\boldsymbol{c}_d) \propto \exp \mathbb{E}_{\boldsymbol{\pi}_d\sim q(\boldsymbol{\pi}_d)}\left[\lambda \boldsymbol{c}_d^\top \ln \boldsymbol{\pi}_d\right] = \exp\left(\lambda \boldsymbol{c}_d^\top\left(\psi(\hat{\boldsymbol{\alpha}}_d) - \psi(\sum_k \hat{\alpha}_{dk})\right)\right) \tag{17}$$

Thus, the update for $\hat{\boldsymbol{p}}_d$ becomes:

$$\hat{\boldsymbol{p}}_d = \frac{\exp\left(\lambda\psi(\hat{\boldsymbol{\alpha}}_d)\right)}{\sum_k \exp\left(\lambda\psi(\hat{\alpha}_{dk})\right)}. \tag{18}$$

**4.** To find the updates for the neural network parameters, we now fix $\hat{\boldsymbol{p}}_{dn}$ and search for $\hat{\boldsymbol{\theta}}$ that maximises the following:

$$
\begin{aligned}
q(\boldsymbol{\theta}) &\propto \exp \mathbb{E}_{\boldsymbol{k}_{dn}\sim q(\boldsymbol{k}_{dn})}\left[r(\boldsymbol{\theta}) + \sum_{dn} \boldsymbol{k}_{dn}^\top g(\boldsymbol{x}_{dn},\boldsymbol{\theta})\right] \\
&= \exp\left(r(\boldsymbol{\theta}) + \sum_{dn} \mathbb{E}_{q(\boldsymbol{k}_{dn})}\left[\boldsymbol{k}_{dn}^\top\right] g(\boldsymbol{x}_{dn},\boldsymbol{\theta})\right) \\
&= \exp\left(r(\boldsymbol{\theta}) + \sum_{dn} \hat{\boldsymbol{p}}_{dn}^\top g(\boldsymbol{x}_{dn},\boldsymbol{\theta})\right)
\end{aligned}
\tag{19}
$$

## Appendix D

In the following, we describe further approximations which will make it easier to implement our approach in practice. In particular, we would like to be able to update $\hat{\boldsymbol{\alpha}}_d$ and $\hat{\boldsymbol{\theta}}$ after observing just a single item (or a mini batch of data points) and not have to process large collections of items or entire datasets in each iteration.

For every document, we keep an estimate of the expected topic counts from which we can calculate $\hat{\boldsymbol{\alpha}}_d$ in Eq. 14:

$$\bar{\boldsymbol{p}}_d = \sum_n \hat{\boldsymbol{p}}_{dn}. \tag{20}$$

Assume that at training step $t$ we update $\hat{\boldsymbol{p}}_{dn}$, then we would also want to perform the following update:

$$\bar{\boldsymbol{p}}_d^t = \bar{\boldsymbol{p}}_d^{t-1} + \hat{\boldsymbol{p}}_{dn}^t - \hat{\boldsymbol{p}}_{dn}^{t-1} \tag{21}$$

Unfortunately, this requires keeping in memory the previous value of $\hat{p}_{dn}$, which is too costly for datasets containing a lot of items. Instead, in every update of $\bar{p}_d$ we subtract an equal fraction of all $\hat{p}_{dn}$:

$$\bar{p}_d^t = \bar{p}_d^{t-1} + \hat{p}_{dn}^t - \bar{p}_d^{t-1}/N_d. \tag{22}$$

Using Jensen's inequality, let us bring the regularizer into a form which is more amenable to stochastic approximation:

$$\begin{aligned}
r(\boldsymbol{\theta}, \boldsymbol{x}) &= \gamma \cdot \mathbf{1}^\top \ln \sum_{dn} \exp g(\boldsymbol{x}_{dn}, \boldsymbol{\theta}) \\
&= \gamma \sum_k \ln \sum_{dn} \frac{r_{dnk}}{r_{dnk}} \exp g_k(\boldsymbol{x}_{dn}, \boldsymbol{\theta}) \\
&\geq \gamma \sum_k \sum_{dn} r_{dnk} \ln \frac{\exp g_k(\boldsymbol{x}_{dn}, \boldsymbol{\theta})}{r_{dnk}}
\end{aligned} \tag{23}$$

Eq. 23 provides us with a lower bound on the regularizer for arbitrary $r_{dnk}$ with $\sum_{dn} r_{dnk} = 1$ for every $k$. The bound is tight when

$$r_{dnk} = \frac{\exp g_k(\boldsymbol{x}_{dn}, \boldsymbol{\theta})}{\sum_{dn} \exp g_k(\boldsymbol{x}_{dn}, \boldsymbol{\theta})}. \tag{24}$$

Intuitively, $r_{dnk}$ is the responsibility of a word $\boldsymbol{x}_{dn}$ for topic $k$. If it is large, a word is associated more strongly with a topic than other words in the dataset.

With the approximation to the regularizer, we obtain a simpler form for the loss function of $\hat{\boldsymbol{\theta}}$:

$$\begin{aligned}
-\ell(\hat{\boldsymbol{\theta}}) &= \sum_{dn} \hat{p}_{dn}^\top g(\boldsymbol{x}_{dn}, \hat{\boldsymbol{\theta}}) + \gamma \cdot \mathbf{1}^\top \ln \sum_{dn} \exp g(\boldsymbol{x}_{dn}, \hat{\boldsymbol{\theta}}) \\
&\geq \sum_{dn} \hat{p}_{dn}^\top g(\boldsymbol{x}_{dn}, \hat{\boldsymbol{\theta}}) + \gamma \sum_{dn} r_{dn}^\top g(\boldsymbol{x}_{dn}, \hat{\boldsymbol{\theta}}) - \gamma \sum_{dn} r_{dn}^\top \ln r_{dn} \\
&= \sum_{dn} \hat{p}_{dn}^\top g(\boldsymbol{x}_{dn}, \hat{\boldsymbol{\theta}}) + \gamma \sum_{dn} r_{dn}^\top g(\boldsymbol{x}_{dn}, \hat{\boldsymbol{\theta}}) + \text{const}
\end{aligned} \tag{25}$$

Here, we assumed that $r_{dnk}$ is computed based on a fixed value of neural network parameters and thus is a constant with respect to $\boldsymbol{\theta}$. If $\boldsymbol{r}_{dn}$ is such that the bound in Eq. 23 is tight, then the gradient at $\hat{\boldsymbol{\theta}}$ will be the same as for the original loss.

Unfortunately, the denominator in Eq. 24 depends on the entire dataset. We therefore propose to estimate the denominator with a running average. If at training step $t$ we observe $\boldsymbol{x}_{dn}$, our estimate $\bar{r}$ is updated as follows:

$$\hat{r}_{dn}^* = \exp g(\boldsymbol{x}_{dn}, \hat{\boldsymbol{\theta}}), \qquad \bar{r}_{t+1} = \bar{r}_t + \varepsilon \left( M \hat{r}_{dn}^* - \bar{r}_t \right), \tag{26}$$

where $M = \sum_d N_d$ is the number of words in the document. With this we can estimate responsibilities as $\hat{r}_{dn} = \hat{r}_{dn}^*/\bar{r}$, and stochastically approximate the loss with

$$-\ell(\hat{\boldsymbol{\theta}})/M \approx \hat{p}_{dn}^\top g(\boldsymbol{x}_{dn}, \hat{\boldsymbol{\theta}}) + \gamma \cdot \hat{r}_{dn}^\top g(\boldsymbol{x}_{dn}, \hat{\boldsymbol{\theta}}). \tag{27}$$

This corresponds to a cross-entropy loss, where the targets of the neural network are given by $\hat{p}_{dn} + \gamma \cdot \hat{r}_{dn}$. The full algorithm for stochastic training of logistic LDA (supervised version) is given in Algorithm 1. An inference procedure which can be used at test time is given in Algorithm 2.

**Algorithm 1** Training step given a single word $\boldsymbol{x}_{dn}$ and a document label $\boldsymbol{c}_d$. Before training, $\bar{\boldsymbol{p}}_d = 0$ and $\bar{\boldsymbol{r}} = \frac{M}{K}\boldsymbol{1}$. $N_d$ is the number of words in each document and $M$ is the total number of words in the dataset.

**Require:** $\boldsymbol{x}_{dn}, \mathbf{c}_d, \hat{\boldsymbol{\theta}}, \bar{\boldsymbol{p}}_d, \bar{\boldsymbol{r}}, N_d, M$

  % Compute logits and fix their values
  $\boldsymbol{f}_{dn} \leftarrow f(\boldsymbol{x}_{dn}, \hat{\boldsymbol{\theta}})$

  % Compute beliefs over latent variables
  $\hat{\boldsymbol{\alpha}}_d \leftarrow \boldsymbol{\alpha} + \bar{\boldsymbol{p}}_d + \lambda \boldsymbol{c}_d$
  $\hat{\boldsymbol{p}}_{dn} \leftarrow \text{softmax}\left(\boldsymbol{f}_{dn} + \psi(\hat{\boldsymbol{\alpha}}_d)\right)$

  % Compute responsibilities
  $\hat{\boldsymbol{r}}_{dn}^* \leftarrow \text{softmax}\left(\boldsymbol{f}_{dn}\right)$
  $\hat{\boldsymbol{r}}_{dn} \leftarrow \hat{\boldsymbol{r}}_{dn}^* / \bar{\boldsymbol{r}}$

  % Update neural network's parameters
  $\hat{\boldsymbol{\theta}} \leftarrow \hat{\boldsymbol{\theta}} - \varepsilon \nabla_\theta \text{crossent}\left(\hat{\boldsymbol{p}}_{dn} + \gamma \cdot \hat{\boldsymbol{r}}_{dn}, f(\boldsymbol{x}_{dn}, \hat{\boldsymbol{\theta}})\right)$

  % Update statistics
  $\bar{\boldsymbol{p}}_d \leftarrow \bar{\boldsymbol{p}}_d + \hat{\boldsymbol{p}}_{dn} - \bar{\boldsymbol{p}}_d / N_d$
  $\bar{\boldsymbol{r}} \leftarrow \boldsymbol{r} + \hat{\boldsymbol{r}}_{dn}^* - \bar{\boldsymbol{r}} / M$

  **return** $\hat{\boldsymbol{\theta}}, \bar{\boldsymbol{p}}_d, \bar{\boldsymbol{r}}$

**Algorithm 2** Stochastic inference, estimating topic proportions by looking at words one by one. This algorithm is expected to be called many times with words sampled from the corpus.

**Require:** $\boldsymbol{x}_{dn}, \bar{\boldsymbol{p}}_d$
  % Beliefs over class labels
  $\hat{\boldsymbol{p}}_d \leftarrow \boldsymbol{1}/K$

  % Beliefs over topic proportions
  **for** $i$ from 1 to $N_{\text{iter}}$ **do**
    $\hat{\boldsymbol{\alpha}}_d \leftarrow \boldsymbol{\alpha} + \bar{\boldsymbol{p}}_d + \lambda \hat{\boldsymbol{p}}_d$
    $\hat{\boldsymbol{p}}_d \leftarrow \text{softmax}(\lambda \psi(\hat{\boldsymbol{\alpha}}_d))$
  **end for**

  % Make prediction for word
  $\hat{\boldsymbol{p}}_{dn} \leftarrow \text{softmax}\left(f(\boldsymbol{x}_{dn}, \hat{\boldsymbol{\theta}}) + \psi(\hat{\boldsymbol{\alpha}}_d)\right)$

  % Update document's topic counts
  $\bar{\boldsymbol{p}}_d \leftarrow \bar{\boldsymbol{p}}_d + \hat{\boldsymbol{p}}_{dn} - \bar{\boldsymbol{p}}_d / N_d$

  **return** $\bar{\boldsymbol{p}}_d, \hat{\boldsymbol{p}}_d, \hat{\boldsymbol{p}}_{dn}$

# Appendix E

For supervised models, we used random search across the following range of hyperparameters:

- number of hidden units in up to three hidden layers:
  128, 256, 512, 1024, [1024 512], [512 256], [256 128], [512 256 128]
- initial learning rate: 0.001, 0.003
- exponential learning rate decay every 2000 iterations: 0.5 to 0.9
- batch size: 8 to 128
- number of training steps: 50000 to 200000
- Dirichlet prior parameter ($\alpha$): 0.5 to 10

For models trained with a mean-field variational inference (Section 5.1 in the main text):

- $\lambda$: 10.0 to 8000.0
- $\gamma$: 0.1 to 1000.0
- number of variational iterations ($N_{\text{iter}}$): 1 to 5

For models trained with the empirical loss (Section 5.2 in the main text):

- $\lambda$: 1.0
- number of variational iterations ($N_{\text{iter}}$): 1 to 20

**Twitter**

The Twitter model was trained with a stochastic approximation of mean-field variational inference as given in Algorithm 1 for approximately 20 minutes on a CPU. For all models evaluated in Table 1 of the main text, we randomly sampled 100 sets of hyperameters. A single model was chosen based on a validation set and evaluated on the test set. The model which achieved 38.7% on the author-level community prediction used the following hyperparameters:

- number of hidden units: [256, 128]
  - initial learning rate: 0.0005
  - exponential learning rate decay every 2000 iterations: 0.8
  - batch size (number of items): 64
  - number of training iterations: 60000
  - $\lambda$: 2000.0
  - $\alpha$: 0.5
  - $\gamma$: 0.25

**Pinterest**

The Pinterest model was trained with a stochastic approximation of mean-field variational inference as given in Algorithm 1 for 5 hours on a CPU with the following hyperparameters:

  - number of hidden units: [256, 128]
  - initial learning rate: 0.001
  - exponential learning rate decay every 2000 iterations: 0.8
  - batch size (number of items): 8
  - number of training iterations: 50000
  - $\lambda$: 8000.0
  - $\alpha$: 5.0
  - $\gamma$: 20.0

The unsupervised logistic LDA model for Pinterest had quite a different range of suitable hyperparameters. We used the model with the following hyperparameters:

  - number of topics: 14
  - number of hidden units: [128]
  - initial learning rate: 0.0001
  - exponential learning rate decay every 2000 iterations: 0.6
  - batch size (number of items): 4
  - number of training iterations: 60000
  - $\lambda$: 1.
  - $\alpha$: 1.0
  - $\gamma$: 1000.0

**20-Newsgroups**

The logistic LDA model that achieves 15.6% error rate on 20-Newsgroups was trained for 11h on a CPU with the following hyperparameters:

  - number of hidden units: [1024 512]
  - initial learning rate: 0.001
  - exponential learning rate decay every 2000 iterations: 0.7
  - number of documents in a batch: 16
  - number of training iterations: 100000
  - number of variational iterations ($N_{\text{iter}}$): 20
  - $\alpha$, $\lambda$: 1.

For the unsupervised logistic LDA trained with a general mean-field VI objective, the range of suitable hyperparameters is quite different. For example, we noticed that it requires lower learning rates and stronger regularization in the form of $\gamma$ from Eq. 27. Thanks to the stochastic approximations, the unsupervised model takes minutes to train on a CPU. To produce the resutls in Table 3 from the main text and tables in Appendix H, we used logistic LDA with the following hyperparameters:

- number of hidden units: 128
- initial learning rate: 0.0005
- exponential learning rate decay every 2000 iterations: 0.8
- number of documents in a batch: 16
- number of training iterations: 50000
- $\alpha$: 5.
- $\lambda$: 1.
- $\gamma$: 100.

## Appendix F

**Twitter**

For the training data, we first collected English tweets from $200\,000$ Twitter users with more than 5,000 followers. The creation dates of the tweets ranged from October 8 to October 10, 2018. Tweets with less than 3 tokens were ommited. Authors were clustered and assigned to communities based on their follower graph. The communities were manually annotated based on a sample of of tweets from the community. We retained 300 communities containing 66,455 authors and discarded the rest, leaving a dataset of 1.45 million tweets.

The test data was generated by sampling 30,000 tweets created between July 1 and August 1, 2018. As for the training data, tweets with fewer than 3 tokens were filtered out. Each tweet is manually annotated with a topic from a taxonomy of over 400 topics. After only retaining tweets corresponding to one of the 300 communities, we were left with 18,864 tweets. For each of the 18,765 authors of these tweets, we sampled a number of additional unlabeled tweets, yielding a total of 3.14 million tweets.

Tweet text was preprocessed by removing accents, replacing certain characters (e.g., "æ" becomes "ae"), and lowercasing words, but no stemming was applied.

**Pinterest**

The Pinterest dataset [5] contains around 46,000 boards divided almost equally among 468 categories. We noticed that board labels are often noisy and sometimes incorrect. Therefore, we manually selected a subset of 90 categories, which we merged into 14 broader classes: food, animals, women_fashion, men_fashion, garden, architecture, wedding, cars, hair, home, travel, DIY, fitness, and fashion. This resulted in a set of 8,989 boards, which we splitted into 7,191 and 1,798 boards for training and testing. On average, every board contains 58 pinned images. We only kept images with a height to width ratio between 0.4 and 2.0, and rescaled every image to $224 \times 224$ pixels.

**20-Newsgroups**

We used the preprocessed 'all-terms' version of 20 Newsgroups [3], where attachments, PGP keys, duplicates and empty messages were removed. This version was chosen for our results to be comparable with the results of Dai and Le [4]. Note that scores can differ greatly when using other variants of 20 Newsgroups, e.g., versions with headers, footers or quotes removed. Unfortunately, the preprocessing done by Cardoso-Cachopo [3] also removed special characters, punctuation, capitalization and numbers. This prevented us to restore sentences and, for instance, reduce the noise coming from mishandling contractions. In order for us to use GloVe word embeddings, we further processed the documents by applying tokenization and stemming.

# Appendix G

Figure 1: Sample of pins from test Pinterest boards covering most of the topics. Board labels and predictions are on the left. Pins predictions and probabilities from logistic LDA and MLP are given below every image. There are some examples where MLP outputs seemingly improper predictions, e.g. 'food' in the top row, due to its lack of context. The same predictions we would get from our model if we let $\hat{\boldsymbol{p}}_{dn} \propto f(\boldsymbol{x}_{dn}, \boldsymbol{\theta})$ instead of $\hat{\boldsymbol{p}}_{dn} \propto f(\boldsymbol{x}_{dn}, \boldsymbol{\theta}) + \psi(\hat{\boldsymbol{\alpha}}_d)$. Thus, having a context-dependent bias in the form of $\hat{\boldsymbol{\alpha}}_d$ leads to more meaningful predictions.

## Appendix H

An unsupervised logistic LDA trained on the Pinterest data discovers topics as illustrated in Figure 2. Here, we classified every pin as belonging to one of the 14 topics, sorted the pins according to $p(\boldsymbol{k} \mid \boldsymbol{x})$, and selected the top-9 pins per topic. The occasional duplicates come from the fact that the same image can be pinned multiple times.

Figure 2: Examples of nine topics discovered by unsupervised logistic LDA in the Pinterest dataset.

## Appendix I

In the unsupervised 20-Newsgroups experiments, we used the vocabulary provided by Srivastava and Sutton [10]. Miao et al. [8] used identical settings. For direct comparison, we also chose the same number of topics, namely 50. We cannot use perplexity or topic coherence to compare logistic LDA with ProdLDA [10] and GSM Miao et al. [8] since our model does not model the distribution of inputs needed for those metrics. Therefore, we can only qualitatively evaluate the models by inspecting the topics. For ProdLDA [10] and GSM Miao et al. [8], a topic is a distribution over words, so the top-10 words in Table 3 and Table 4 are sorted based on $p(\boldsymbol{x} \mid \boldsymbol{k})$. On the other hand, logistic LDA models the probability of topics given a word $p(\boldsymbol{k} \mid \boldsymbol{x})$. Thus, we can classify every word in the

vocabulary as belonging to some topic and sort the words within each topic according to the topic probability. In Table 1 and Table2 we list top-10 of those words.

Table 1: Selected topics discovered by unsupervised logistic LDA represented by top-10 words

**1** bmw, motor, car, honda, motorcycle, auto, mg, engine, ford, bike
**2** christianity, prophet, atheist, religion, holy, scripture, biblical, catholic, homosexual, religious, atheist
**3** spacecraft, orbit, probe, ship, satellite, rocket, surface, shipping, moon, launch
**4** user, computer, microsoft, monitor,programmer, electronic, processing, data, app, systems
**5** congress, administration, economic, accord, trade, criminal, seriously, fight, responsible, future

Table 2: Five *randomly* selected topics discovered by logistic LDA represented by top-10 words

**1** homicide, city, south, area, national, north, department, since, highest, disease
**2** marry, mother, father, love, lord, mary, soul, hell, author, life
**3** angeles, philadelphia, chicago, vancouver, toronto, pittsburgh, colorado, detroit, university, nhl
**4** science, editor, english, letter, journal, art, together, first, award, study
**5** gif, jpeg, pixel, font, postscript, unix, email, server, binary, directory

Table 3: Five *randomly* selected topics discovered by ProdLDA [10] represented by top-10 words

**1** motherboard, meg, printer, quadra, hd, windows, processor, vga, mhz, connector
**2** armenian, genocide, turks, turkish, muslim, massacre, turkey, armenians, armenia, greek
**3** voltage, nec, outlet, circuit, cable, wiring, wire, panel, motor, install
**4** season, nhl, team, hockey, playoff, puck, league, flyers, defensive, player
**5** israel, israeli, lebanese, arab, lebanon, arabs, civilian, territory, palestinian, militia

Table 4: Selected topics discovered by GSM [8] represented by top-10 words

**1** space, satellite, april, sequence, launch, president, station, radar, training, committee
**2** god, atheism, exist, atheist, moral, existence, marriage, system, parent, murder
**3** encryption, device, technology, protect, americans, chip, use, privacy, industry, enforcement
**4** player, hall, defensive, team, average, career, league, play, bob, year
**5** science, theory, scientific, universe, experiment, observation, evidence, exist, god, mistake

To quantitatively evaluate the quality of discovered topics, we computed the NPMI topic coherence scores [7] of our model and compared them to the scores published by Miao et al. [8]. The results are given in the table below. We find that logistic LDA outperforms other topic models, but performs similar to document models which represent topics implicitly and are thus more difficult to interpret.

|  | $K = 50$ | $K = 200$ | Explicit topics | Generative |
|---|---|---|---|---|
| LogisticLDA | 0.215 | 0.179 | ✓ | × |
| OnlineLDA [6] | 0.131 | 0.112 | ✓ | ✓ |
| NVDM [9] | 0.186 | 0.157 | × | ✓ |
| NVLDA [10] | 0.110 | 0.110 | ✓ | ✓ |
| ProdLDA [10] | 0.240 | 0.190 | × | ✓ |