[Reviews · NeurIPS 2019]

Reviewer 1



ORIGINALITY The idea of having a discriminative version of LDA, analogous to logistic regression, is interesting. This idea is carried out quite well with the logistic LDA, its inference algorithm, and classification results using various datasets. QUALITY One concern I have is with comparisons with supervised LDA models, such as sLDA, discLDA, or LLDA. I realize these are mentioned in the beginning of the paper, and authors may have felt they are not as relevant, as they are not discriminative models, but I feel that readers would natural wonder about this, and authors should compare them, not necessarily empirically (thought that would be helpful). Another question I had was about topics being coherent. This paper (and supplementary PDF) shows example topics, both for images and text, but the more accepted evaluation is to actually compute topic coherence, which perhaps cannot be done for images but certainly for documents. The paper says "We find that logistic LDA is able to learn coherent topics in an unsupervised manner.", but I feel this is not supported with sufficient evidence. CLARITY This paper is very well written, and it is mostly very clear. However, I had trouble understanding the following few things: LDA discovers, for each document, a distribution over topics. Does logistic LDA also assign multiple topics (and a distribution over them) for each item? If not, I think this paper should make that clear and discuss this limitation. Perhaps related to this point, for evaluation of tweets, the papers says that when an author belongs to multiple communities, one is chosen at random. What would this mean in terms of the author classification results? Lastly, in the tweet classification, I did not fully understand what is meant by the sentence "For LDA, we extended the open source implementation of Theis and Hoffman [36] to depend on the label in the same manner as logistic LDA." I am pretty familiar with the topic modeling literature, and I think this would need more explanation. Miscellaneous question/comment -- Authors mention that one of their contributions is a new dataset of annotated tweets. As far as I know, Twitter does not allow distributing tweets that researchers collect. Please make sure and describe exactly how these data will be distributed. ** Post-author response ** I appreciate the authors responding to the questions and concerns in my review. I am happy with the response and raised my score accordingly.

Reviewer 2



I have mixed feelings about this paper. On the bright side, I like the idea of relaxing the (sometimes strict) assumptions underlying topic models such as LDA. The differenciable g functions act as a comparator between the topic distribution over words p(w|z) and the vector representation of w. It reminds me some recent works that combine topic models and word embedding (see [1]). It is an interesting way to embed specific preprocessing, such as convolutionnal layers for dealing with images. On the other side I'm not convinced that this model is just "another view" of the classic LDA. Let's take an example: the authors use a Dirichlet prior to "mimick" the behavior or LDA when generating \pi_d (what I usually call \theta_d), but it's not really motivated here. This prior is usually chosen for calculation purpose, using the conjugacy between distributions. Why following the same path here? Generally speaking, I face difficulties in fully understanding the analogy with LDA. It *looks* similar but I still think we loose the beauty and fundations of probabilistic models. The paper is probably too short with (too) many appendices, and it is hard to follow the multiple derivations (e.g., parameters inference). The authors chose to add an extension to let their model deal with observed classes. However there is a huge literature for integrating supervision to LDA. sLDA isn't the only model (see for instance labeled LDA [2]). Besides, the assumption is that there is a one-to-one relation between topics and classes. I'm not fully convinced by the experiments that it is a fruitful assumption, which is annoying with the title chosen by the authors. Therefore I suggest to remove this part of the contribution (and fin another title), or to submit to a journal to have room for giving a clear presentation of the work. Finally I see pros and cons for the experimental section. It's definitively a good idea to vary the kind of datasets, but it also gives arguments against here and there. For instance: - Several topic models have been proposed to deal with short texts, in particular posted on Twitter. See for instance the Biterm topic model [3]. - The authors use two different evaluation measures for classification (see Tables 1 and 2). Why? - I'm highly surprised that we can write that learning a model for 20NewsGroups (~20,000 instances) in 11 hours is fast! I'm highly confident that we can train classification models faster with competitive results. - I'm hardly convinced by the top words given in Table 3 (and in the appendices). Recent papers use topic coherence measures, such as the ones based on NPMI (see [4]). I spotted some typos, such as: - "Table 11" (line 239) - "embeddigns" (line 269) === UPDATE: I've carefully read the other reviews and authors' response. It wasn't enough to fully convince me on a couple of points (e.g., complexity in time, topic coherence that "will be included"). However I've changed my overall score from 6 (weak accept) to 7 (accept). === References [1] Das, R., Zaheer, M., & Dyer, C. (2015, July). Gaussian lda for topic models with word embeddings. In Proceedings of the 53rd Annual Meeting of the Association for Computational Linguistics and the 7th International Joint Conference on Natural Language Processing (Volume 1: Long Papers) (pp. 795-804). [2] Ramage, D., Hall, D., Nallapati, R., & Manning, C. D. (2009, August). Labeled LDA: A supervised topic model for credit attribution in multi-labeled corpora. In Proceedings of the 2009 Conference on Empirical Methods in Natural Language Processing: Volume 1-Volume 1 (pp. 248-256). Association for Computational Linguistics. [3] Yan, X., Guo, J., Lan, Y., & Cheng, X. (2013, May). A biterm topic model for short texts. In Proceedings of the 22nd international conference on World Wide Web (pp. 1445-1456). ACM. [4] Röder, M., Both, A., & Hinneburg, A. (2015, February). Exploring the space of topic coherence measures. In Proceedings of the eighth ACM international conference on Web search and data mining (pp. 399-408). ACM.

Reviewer 3



The paper was expertly written which introduces an interesting discriminative variant of LDA. I think this work will be a nice addition to the crowded literature of topic modeling. Here are some of my additional thoughts: - One of the many advantages of LDA is that it is a building block for many topic modeling extensions, as described Section 2.2 “A zoo of topic models” in the paper. I wondering with this discriminative variant, how easy/difficult it is to modify Logistic LDA to achieve these extensions. - It also would be helpful if the paper discusses the scalability of the two algorithms proposed to train Logistic LDA

[Author Response · NeurIPS 2019]

We thank the reviewers for their time and careful reviews. Please find our responses below.

**General.** We will update the paper with NPMI topic coherence scores and compare to results published by Miao et al. (2018). We find that logistic LDA outperforms other *topic models*, but performs similar to *document models* which represent topics implicitly and are thus more difficult to interpret.

|  | $K = 50$ | $K = 200$ | Explicit topics | Generative |
|---|---|---|---|---|
| LogisticLDA | 0.215 | 0.179 | ✓ | × |
| OnlineLDA | 0.131 | 0.112 | ✓ | ✓ |
| NVLDA | 0.110 | 0.110 | ✓ | ✓ |
| NVDM | 0.186 | 0.157 | × | ✓ |
| ProdLDA | 0.240 | 0.190 | × | ✓ |

**Reviewer 1.** The relationship between sLDA and logistic LDA is discussed in Section 3.1 and illustrated in Figure 1. We will add some detail on discLDA and LLDA. As for empirical comparisons with sLDA, we tried to get its open-source code to work on our supervised tasks, but it did not scale to the size of the Twitter dataset. We thus extended another implementation of LDA to be supervised in the same manner as logistic LDA (Eq.6). What we call "LDA" in Table 1 is in fact this supervised extension. This is related to your comment regarding the necessity to better explain how we extended the implementation of Theis and Hoffman (2015). We appreciate this remark and will improve our paper on this front.

Like LDA, logistic LDA indeed discovers a distribution over topic proportions for each document (Eq. 1 and 12), as well as over topics for each item (Eq. 2 and 13). We will make this point more explicit by improving lines 175-178.

Regarding the Twitter dataset, we will release lists of tweet IDs annotated by topic IDs. We will not provide the correspondence between topic IDs and topic names or the tweet text itself to comply with legal policies such as GDPR.

Where authors are annotated with multiple communities/topics in the dataset, we chose one at random. This means the ground-truth label is noisy, limiting the maximum achievable accuracy.

**Reviewer 2.** We are grateful for the provided references and will include them in Section 2.2. We will also add more detail on other supervised extensions.

Regarding the "alternative view" of LDA, we appreciate the feedback and will try to improve its presentation. Note that the assumptions in Eq. 1 to 3 together with Eq. 4 fully characterize LDA, where instead of deriving LDA from priors and likelihoods, we derive LDA from a set of conditional distributions. This view suggests other modifications to the classical view of LDA, and our paper presents one of them. Future work may want to explore alternatives to the Dirichlet assumption, but this is beyond the scope of our paper.

Regarding the loss of beauty and probabilistic foundations, note that probabilistic models do not have to be generative but they are merely sets of assumptions. LDA corresponds to one such set of assumptions and logistic LDA to a modified set of assumptions, which in both cases yield a proper probabilistic model.

In the Twitter and 20-Newsgroups experiments, having a one-to-one mapping from topics to classes seems to be a reasonable assumption, but we agree that in general, more flexibility is desirable. To relax this assumption, we outlined a solution in lines 137-139. In defense of our title, please note that it does not mention that our model is supervised. Logistic LDA is a discriminative topic model in the sense that it does not model the distribution of inputs (see also lines 301-305). Logistic LDA can be unsupervised, supervised or semi-supervised. Regarding the appendix, it may appear lengthy because we included derivations of some known results so that readers do not have to consult multiple sources.

It is indeed incoherent to use *accuracy* in Table 1 and *error rate* in Table 2. We will report accuracy in both tables.

The reported training time of 11 hours was obtained using a CPU. It reduces to less than 1 hour when using a GPU, which is considerably faster compared to LSTM-based models. An SVM is even faster, however, its results are not competitive. We will update the appendix with GPU training time estimates.

**Reviewer 3.** Regarding extensibility, logistic LDA can be modified in the same ways as LDA when the changes target common components of the two models, such as the distribution of topics $\pi_d$. For example, one could replace the Dirichlet with a logistic normal distribution or allow for an unbounded number of topics. Our supervised version of logistic LDA illustrates that adding extra information to the model and using different losses is as easy as in LDA.

We implemented both online and batched versions of training and inference algorithms. In the former case, the updates can be done after seeing a single word or a batch of words from possibly multiple documents. This allows us to process large collections of arbitrary length, which is particularly interesting for the Pinterest and Twitter cases, where more pins or tweets become available. We will add this discussion on scalability to Section 5.1.

[Meta-Review · NeurIPS 2019]

The reviewers all believe this is a solid paper that should be accepted. Reviewer 2 thinks that topic coherence measures would be useful to include, but this isn't a "requirement".